# Expression and Prognostic Value of Chromobox Family Proteins in Esophageal Cancer

**DOI:** 10.3390/genes13091582

**Published:** 2022-09-03

**Authors:** Jin Liu, Haixiang Shen, Xiangliu Chen, Yongfeng Ding, Haiyong Wang, Nong Xu, Lisong Teng

**Affiliations:** 1Department of Surgical Oncology, The First Affiliated Hospital, Zhejiang University School of Medicine, Hangzhou 310030, China; 2Department of Urology, The First Affiliated Hospital, Zhejiang University School of Medicine, Hangzhou 310030, China; 3Department of Medical Oncology, The First Affiliated Hospital, Zhejiang University School of Medicine, Hangzhou 310030, China

**Keywords:** esophageal cancer (EC), Chromobox (CBX), biomarker

## Abstract

Background: Esophageal cancer (EC) is one of the most common human malignant tumors worldwide. Chromobox (CBX) family proteins are significant components of epigenetic regulatory complexes. It is reported that CBXs play critical roles in the oncogenesis and development of various tumors. Nonetheless, their functions and specific roles in EC remain vague and obscure. Methods and Materials: We used multiple bioinformatics tools, including Oncomine, Gene Expression Profiling Interactive Analysis 2 (GEPIA2), UALCAN, Kaplan–Meier plotter, cBioPortal, Metascape, TIMER2 and TISIDB, to investigate the expression profile, gene alterations and prognostic roles of CBX family proteins, as well as their association with clinicopathologic parameters, immune cells and immune regulators. In addition, RT-qPCR, Western blot, CCK8, colony formation, wound healing and transwell assays were performed to investigate the biological functions of CBX3 in EC cells. Results: CBX3 and CBX5 were overexpressed in EC compared to normal tissues. Survival analysis revealed that high expression of CBX1 predicted worse disease-free survival (DFS) in EC patients. Functionally, CBXs might participate in mismatch repair, spliceosome, cell cycle, the Fanconi anemia pathway, tight junction, the mRNA surveillance pathway and the Hippo signaling pathway in EC development. Furthermore, CBXs were related to distinct immune cells infiltration and immune regulators. Additionally, depletion of CBX3 inhibited the proliferation, migration and invasion abilities of EC cells. Conclusions: Our study comprehensively investigated the expression pattern, prognostic value, and gene alterations of CBXs in EC, as well as their relationships with clinicopathologic variables, immune cells infiltration and immune regulators. These results suggested that CBX family proteins, especially CBX3, might be potential biomarkers in the progression of EC.

## 1. Introduction

According to GLOBOCAN 2020 report, there are 604,000 newly diagnosed patients with esophageal cancer (EC) and 544,000 deaths annually, meaning that it is ranked seventh in incidence and sixth in mortality [1]. Esophageal cancer consists of two major different subtypes, esophageal squamous cell carcinoma (ESCC) and esophageal adenocarcinoma (EAC), which share the same anatomical site, differ in epidemiology and are distinct in pathology [2]. Esophagectomy, radiotherapy and chemotherapy are still important cornerstones in EC treatment. Targeted therapy and immunotherapy also improve the survival rate of EC patients [3,4]. For immunotherapy, several phase III studies have provided evidence that immune checkpoint inhibitors in combination with or without chemotherapy improved the median overall survival (OS) of EC patients in first-line or second-line treatment [5,6,7], whereas the 5-year survival rate of EC patients remains low at 30–40% owing to the advanced stage of EC at initial diagnosis, tumor heterogeneity, drug resistance and gene alterations during treatment courses [1]. It has been reported that the early diagnosis and management of EC could improve the outcomes and decrease mortality [8]. Recently, some novel blood biomarkers including miRNAs, lncRNAs, and metabolite and serum antibodies have been reported to be potentially applied in the screening and early detection of EC, shedding new light on the biomarkers for EC management [9]. Collectively, elucidating the precise mechanism involved in EC progression and identifying credible biomarkers to assist clinical treatment is an urgent task.

The epigenetic regulatory mechanisms such as DNA and RNA methylation play vital roles in malignancies [10]. As epigenetic regulatory complexes, polycomb group (PcG) complexes consist of two best characterized forms, polycomb repressive complexes 1 and 2 (PRC1 and PRC2) [11]. Chromobox (CBX) family proteins are typical components in PRC1, mainly responsible for targeting PRC1 to the chromatin. There are eight different proteins among CBX family in total, and they all contain a single N-terminal chromodomain. Among them, CBX1/3/5, also known as heterochromatin protein 1 β (HP1β), HP1γ, and HP1α, respectively, belong to the HP1 group, and CBX2/4/6/7/8 belong to the polycomb group based on the evidence that they all contain a C-terminal polycomb repressor box [12,13]. 

Prior studies have revealed that CBX family proteins are involved in both physiology and pathology processes. Additionally, the roles of CBXs in cancer have been increasingly examined recently. It was found that the microRNA-758-3p–CBX5 axis promoted tumor progression in gastric cancer (GC) [14]. CBX7 can act as an oncogene and is positively associated with clinicopathologic features in patients with GC, including clinical stage, lymph node metastasis and age [15]. Moreover, CBX7 participated in the AKT pathway and contributed to maintaining the stemness of GC cells [16]. CBX family proteins were also involved in the progression of hepatocellular carcinoma (HCC) [17], breast cancer [18], lung cancer [19], colon cancer [20] and pancreatic cancer [21]. However, their roles in EC remain to be elucidated. In the present study, we systematically explored the expression profiles, gene alterations and prognostic roles of CBX family proteins in EC, as well as their association with clinicopathologic parameters, immune cells infiltration and immune regulators. In addition, we also explored the underlying mechanisms and pathways of how CBXs might participate in the progression of EC.

## 2. Methods and Materials

### 2.1. Oncomine

Oncomine (www.oncomine.org, accessed on 1 September 2021) acts as an online cancer database which provides public microarray information for more than 20 types of cancers [22]. In this study, we used Oncomine to highlight the expression profile of CBXs by comparing the mRNA expression level between the EC and normal tissues. The thresholds including *p* < 0.05, fold-change > 2 and gene rank in the top 10% were considered statistically significant. 

### 2.2. GEPIA2

Gene Expression Profiling Interactive Analysis 2 (GEPIA2, http://gepia2.cancer-pku.cn/, accessed on 1 September 2021) is a web server helping researchers to conduct gene expression profiling between cancer and normal tissues, as well as interactive analyses based on RNA-seq datasets from the Cancer Genome Atlas (TCGA) and Genotype-Tissue Expression (GTEx) projects [23]. In our study, we used GEPIA2 to determine the distinct expression of CBXs between the EC and normal samples. In addition, the relationships between the CBXs and clinical tumor stage of EC, OS and disease-free survival (DFS) in EC patients were analyzed through GEPIA2. 

### 2.3. UALCAN

UALCAN (http://ualcan.path.uab.edu/, accessed on 1 September 2021) provides an approach to TCGA data and clinical data, allowing users to identify gene expression and correlation, methylation, and survival analysis of genes of interest [24]. We applied UALCAN to validate the expression of CBXs in ESCC and EAC, and revealed the relationships between CBXs and clinicopathologic parameters (tumor pathological types, nodal metastatic status) in EC patients. *p* < 0.05 indicated statistical significance.

### 2.4. Kaplan–Meier Plotter

Kaplan–Meier plotter (http://kmplot.com/analysis/, accessed on 1 September 2021) provides access to analyzing the correlation between genes and survival in 21 types of cancers [25]. In the present study, we used the Kaplan–Meier plotter to perform survival analysis based on the level of CBXs.

### 2.5. cBioPortal

cBioPortal (https://www.cbioportal.org/, accessed on 20 September 2021) is an online open-access web server used to analyze genomics data of cancer [26]. We used cBioPortal to construct a multidimensional frame of genetic alterations of CBXs in EC. 

### 2.6. Metascape

Metascape (http://metascape.org, accessed on 20 September 2021) is a web-based tool to analyze and interpret OMICs-based studies [27]. In this study, we performed Gene Ontology (GO) and Kyoto Encyclopedia of Genes and Genomes (KEGG) pathway enrichment analysis of CBXs in EC via the “Custom Analysis” model.

### 2.7. TIMER2

TIMER2 (https://cistrome.shinyapps.io/timer/, accessed on 30 September 2021) allows users to evaluate tumor immune infiltration across diverse cancer types and their impact on prognosis [28]. We systematically analyzed the correlation of CBXs with CD8+ T cells, CD4+ T cells, cancer associated fibroblasts (CAFs), myeloid dendritic cells, macrophages, B cells and neutrophils in EC.

### 2.8. TISIDB

TISIDB (http://cis.hku.hk/TISIDB, accessed on 30 September 2021) is a web portal which integrates multiple types of data resources in oncoimmunology, allowing users to comprehensively investigate tumor-immune interactions [29]. We explored the relationships between CBXs and immune regulators in EC with TISIDB at present study.

### 2.9. Cell Culture and Transfection

Human ESCC lines Kyse150, Kyse510, Kyse30, Kyse140 and a normal esophageal cell line HEEC were obtained from the Shanghai Cell Bank of Chinese Academy of Sciences (Shanghai, China) and cultured with RPMI 1640 (Gibco, Rockville, MD, USA) supplemented with 10% fetal bovine serum (FBS, Gibco). Cells were incubated in an incubator (Thermo Fisher, Waltham, MA, USA) with 5% CO_2_ at 37 °C.

The Kyse150 and Kyse510 cells were seeded into the six-well plates. The small interfering RNA (siRNA; GenePharma, Shanghai, China) were transfected with the jetPRIME transfection reagent (Polyplus, Strasbourg, France) according to the manufacturer’s instructions. The siRNA sequences used in the present study were as follows: si-CBX3:5′-GCG TTTCTTAACTCTCAGAAA-3′, and negative control (NC), 5′-TTCTCCGA ACGTGTCACGT-3′. 

### 2.10. RNA Isolation and Real-Time Quantitative PCR

The detailed procedures were conducted as previously described [30]. The primers are as follows: GAPDH-F, 5′-GGAGCGAGATCCCTCCAAAAT-3′; GAPDH-R, 5′-GGCTGTTGTCATACTTCTCATGG-3′; CBX3-F, 5′-TAGATCGACGTGTAGTGAATGGG-3′, CBX3-R, 5′-TGTCTGTGGCACCAATTATTCTT-3′. The target gene expression was calculated by means of the 2^−^^ΔΔCt^ (delta-delta-Ct algorithm) method, with GAPDH as the endogenous reference.

### 2.11. Western Blot Assay

The specific procedures were performed according to a prior study [30]. The primary antibodies used in the present study are as follows: anti-GAPDH (10494-1-AP, Proteintech, Chicago, IL, USA) and anti-CBX3 (11650-2-AP, Proteintech, Chicago, IL, USA). GAPDH was the internal reference.

### 2.12. Colony Formation and Cell Viability Assays

The exact procedures were performed as in a previous study [31].

### 2.13. Wound Healing and Transwell Assays

The detailed processes were performed as in a previous study [32]. The images were collected by the phase-contrast microscope (Olympus) for analysis. 

### 2.14. Statistical Analysis

The data were presented as the means ± SD. Student’s *t*-test or chi-square tests were used to define the statistical significance of two groups. A *p* value < 0.05 was considered of statistical significance.

## 3. Results

### 3.1. Expression Profile of CBX Family Proteins in EC

We firstly used Oncomine to investigate the transcriptional expression of CBXs in EC. As shown in Figure 1, CBXs were differently expressed in 20 types of cancers. That is, CBX1 was significantly overexpressed in cervical cancer, sarcoma, lung cancer, liver cancer, GC and other cancer. For CBX2, upregulation patterns were observed in bladder cancer, breast cancer (BC), colorectal cancer (CRC), GC, lung cancer and other cancer. Additionally, CBX3 was significantly overexpressed in almost all of the 20 cancers showed in the figure, except for liver cancer, pancreatic cancer and leukemia. CBX4 was significantly overexpressed in GC, leukemia and prostate cancer and other cancer. CBX5 was significantly overexpressed in lung cancer, lymphoma, pancreatic cancer and other cancer. With regard to CBX6, many types of cancers harbored low expression, such as brain and CNS cancer, BC and CRC. Additionally, CBX7 was found to be downregulated in most cancers, including BC, cervical cancer, lung cancer and sarcoma. As for CBX8, the current data display that a high expression pattern was observed in most cancers, such as CRC, BC, ovarian cancer and other cancer. For EC, CBX3 and CBX6 were significantly overexpressed in tumor tissues compared to normal tissues. Meanwhile, we used the GEPIA2 database to further verify the mRNA expression of CBXs in EC (Appendix A). We found that both CBX3 and CBX5 were significantly upregulated in EC compared to normal tissues, which was almost consistent with the results obtained from Oncomine. 

### 3.2. Relationships between CBX Family Proteins and Clinicopathologic Variables in EC

According to the analyses in GEPIA2 (Figure 2), we found that CBX1 expression was related to the tumor stage of EC (*p* < 0.05), while no significant correlations were observed for other CBXs (*p* > 0.05). UALCAN database was also applied to verify the expression of CBXs in ESCC and EAC, respectively (Appendix A). We found that CBX1-4 and CBX8 were overexpressed in both ESCC and EAC compared to normal tissues, and overexpression of CBX5 was observed only in ESCC. On the other hand, no difference was observed of CBX6 between EC and normal tissues, but the expression was dramatically distinct in ESCC and EAC tissues. Regard to CBX7, there was no difference observed in EC compared to normal tissues. Subsequently, the relationships between CBXs expression and nodal metastatic status were explored. As shown in Figure 3, EC samples with N1 and N2 stage displayed higher CBX1 compared to samples with N0 stage, while samples with N3 stage expressed lower CBX1 compared to those with N1 stage. Additionally, a high expression of CBX2-4 and CBX8 was detected in N1/2/3 samples compared to normal tissues. Moreover, the expression level of CBX6 was not related to the nodal metastatic status. CBX5 was significantly overexpressed in N1 samples when compared to normal samples, but no significant difference was observed in other nodal metastatic statuses. Of note, CBX7 was downregulated in N3 samples compared to N1 and N2 samples, suggesting that patients with a low expression of CBX7 harbored more lymph node metastasis.

### 3.3. Prognostic Roles of CBX Family Proteins in EC

By analyzing data from GEPIA2, we found that a high expression of CBX1 was related to worse DFS, whereas no significant difference was observed in OS (Figure 4A, *p* < 0.05). For other CBXs, the expression was not significantly associated with either OS or DFS (Figure 4B–H, *p* > 0.05). Considering that pathological types of EC might affect the prognostic value of CBXs in EC, we subsequently used Kaplan–Meier plotter to further determine the prognostic roles of CBXs in ESCC and EAC, respectively. We found that higher expression of CBX4 was correlated with poorer OS compared to the lower group (*p* < 0.05), while the expression of CBX1-3 and CBX5-8 were not significantly related to OS in ESCC patients (*p* > 0.05) (Appendix A). For patients with EAC, a higher expression of CBX3 and CBX8 indicated worse OS, while higher CBX7 expression was associated with better OS (*p* < 0.05, Appendix A).

### 3.4. Genetic Alterations of CBX Family Proteins and the Related Prognostic Value in EC

We used cBioPortal to detect genetic alterations of CBXs in EC. Amplification was the most commonly observed alteration (Figure 5A). As shown in Figure 5B, a total of 1373 samples from 1341 patients were analyzed and 11% among them were altered. We found all of the CBXs harbored diverse genetic alterations including inframe mutation, splice mutation, missense mutation, truncating mutation, amplification and deep deletion. Additionally, the percentage of genetic alterations of CBX1-8 was 1.8%, 2.1%, 4%, 2.1%, 2.3%, 2.3%, 0.6% and 3%, respectively. To evaluate the prognostic significance of genetic changes of CBXs in EC patients, K–M curves were created and no significant relationships between the genetic alterations and OS or DFS were observed (Figure 5C,D, *p* > 0.05). 

### 3.5. GO and KEGG Analysis

To demonstrate the potential functions and pathways of CBXs in EC, we firstly applied cBioPortal to obtain the positively co-expressed genes of each CBXs, among which the top 100 genes of each CBX gene were selected to operate function analysis via Metascape. GO function enrichments including biological processes (BPs), cellular components (CCs) and molecular functions (MPs) were mainly involved in the chromosomal region, DNA repair, mitotic cell cycle phase transition, spindles, DNA replication and intracellular protein-containing complexes (Figure 6A). KEGG analysis demonstrated that CBXs might be involved in mismatch repair, spliceosome, cell cycle, the Fanconi anemia pathway, tight junctions, the mRNA surveillance pathway and the Hippo signaling pathway in EC development (Figure 6B).

### 3.6. Relationships between CBX Family Proteins and Immune Cells Infiltration Level and Immune Regulators in EC

According to the data on TIMER2.0, we found that CBXs were related to the infiltration of various immune cells. That is, CBX1 was positively associated with CD4+ T cells, CAFs, myeloid dendritic cells, macrophages and neutrophils with r = 0.313, 0.448, 0.163, 0.242 and 0.186, respectively, while it was negatively related to CD8+ T cells and B cells with r = −0.27 and −0.162, respectively (Figure 7A). As shown in Figure 7B, CBX2 was positively associated with CD4+ T cells, CAFs, myeloid dendritic cells and neutrophils but negatively associated with CD8+ T cells, macrophages and B cells. CBX3 was positively related to CAF, macrophages and neutrophils, and negatively related to CD8+ T cells, CD4+ T cells, myeloid dendritic cells and B cells (Figure 7C). For CBX4, all seven types of immune cells were positively related to it, except for CAF (Figure 7D). CBX5 was positively related to CD4+ T cells, CAF, myeloid dendritic cells and macrophages, and negatively related to CD8+ T cells, B cells and neutrophils (Figure 7E). As displayed in Figure 7F, only CD8+ T cells and B cells were negatively correlated with CBX6, and the other five types of immune cells were all positively related to CBX6. Similarly, only neutrophils were negatively related to CBX7, while other immune cells had a positive correlation (Figure 7G). In addition, CBX8 was positively related to CD4+ T cells and macrophages but negatively related to the other five types of immune cells (Figure 7H). 

To further explore the roles of CBXs in the regulation of tumor immune microenvironment, the TISIDB database was explored. From the results, we found that CBXs were related to immune regulators (Appendix A). CBX1-6 and CBX8 were all negatively related to immunoinhibitors (Appendix A), immunostimulators (Appendix A) and MHC molecules (Appendix A), while CBX7 was positively associated with the three types of immune regulators.

### 3.7. CBX3 Promotes Cell Proliferation, Migration and Invasion of EC Cells In Vitro

According to the above bioinformatics analyses, we found that CBX3 was overexpressed in almost 20 various types of cancers, including EC. Through the study and the literature review, we found that a high expression of CBX3 might be a biomarker for poor prognosis in HCC, CRC, prostate cancer and osteosarcoma patients [33,34,35,36]. In addition, our study found that high expression of CBX3 was significantly related to worse OS in EAC patients. Considering that CBX3 has a tumor-promoting effect in multiple tumors, we speculated that CBX3 might be involved in EC progression. Given that ESCC accounts for up to 70% of EC across the world and 90% in the region known as the “esophageal cancer belt”, which includes China [2,9], we performed subsequent functional experiments in ESCC cell lines to investigate the role of CBX3. Firstly, when compared to the normal esophageal epithelial cell line HEEC, the upregulation of CBX3 in ESCC cell lines was validated by q-PCR (Figure 8A). Then, we applied targeted siRNAs to knock down the expression of CBX3 in Kyse150 and Kyse510 cell lines. The knockdown efficiency was validated by RT-q-PCR (Figure 8B,C) and Western blot assays (Appendix A). CCK-8 and colony formation indicated that CBX3 knockdown significantly suppressed the proliferation of EC cells (Figure 8D,E). We also performed transwell and wound-healing assays to demonstrate the influence of CBX3 on migration and invasion of EC cells. The data revealed that the migration and invasion abilities of EC cells were significantly impaired in CBX3-silenced cells (Figure 8F–I). 

## 4. Discussion

Accumulative evidence has indicated that CBX family proteins involved in the development of various cancers, such as GC, HCC, CRC, pancreatic cancer, breast cancer and lung cancer [14,15,17,18,19,20,21], while their roles in EC remain to be revealed. Herein, we first demonstrated a comprehensive analysis to explore the roles of CBXs in EC. We found that CBX3 and CBX5 were upregulated in EC, and high CBX3 predicted shorter OS in EAC. Functionally, CBX family proteins might participate in mismatch repair, the spliceosome, the cell cycle pathway in EC. In addition, CBX family proteins were associated with varied immune cell infiltration and immune regulators.

Recent studies have revealed that CBX1 was upregulated in HCC, castration-resistant prostate cancer and breast cancer, and a high expression of CBX1 predicted worse survival outcomes in HCC and shorter recurrence-free survival in BC patients [17,18,37]. However, CBX1 was also reported as being downregulated and contributing to tumor progression in thyroid carcinoma [38]. In the present study, CBX1 was found to be downregulated in EC compared to normal tissues.

Increasing research has reported the roles of CBX3 in cancer development. Prior studies have demonstrated that CBX3 was overexpressed in GC [39], pancreatic cancer [21], HCC [33], osteosarcoma [34], CRC [35], prostate cancer [36] and non-small cell lung cancer (NSCLC) [40]. In addition, high expression of CBX3 might be a biomarker for poor prognosis in HCC [33], CRC [35], prostate cancer [36] and osteosarcoma patients [34]. Mechanically, several miRNAs have been reported to regulate the expression of CBX3 in cancer. In CRC, CBX3 promoted cell proliferation by directly targeting p21 via miR-30a-CBX3 axis [35]. Additionally, miR-139 was verified to upregulate the expression of CBX3 in HCC [41]. Furthermore, as reported in the study by Lin and colleagues, CBX3 exerted pro-tumor function by promoting cell proliferation and the migration of GC cells. Additionally, high expression of CBX3 was inversely associated with the infiltration level of immune cells, as well as immunotherapy response in GC [39]. Similarly, in pancreatic cancer, CBX3 was reported to promote cell proliferation and regulate aerobic glycolysis by suppressing FBP-1 [21]. In our study, we found that CBX3 was upregulated in both ESCC and EAC compared to normal tissues and related to nodal metastasis status. Higher CBX3 was also associated with shorter OS in EAC. Moreover, CBX3 was related to varied immune cells infiltration. Furthermore, the results from in vitro functional experiments revealed that silenced CBX3 impaired the proliferation, migration and invasion of EC cells. Altogether, our results demonstrate that CBX3 was involved in the progression of EC and might be a promising oncogene in EC. 

CBX5 is another widely reported CBX family protein in cancers. Overexpressed in GC tissues and regulated by miR-758-3p, CBX5 promotes invasive biological function in GC cells [14]. CBX5 was also reported to be involved in the SNHG11-miR-2355-5p/CBX5 axis to affect the proliferation and migration of triple-negative breast cancer (TNBC) cells [42]. In addition, interacting with BRD4, CBX5 can inhibit DNA damage response in ovarian cancer [43]. In the present study, we found that CBX5 was overexpressed in EC compared to normal tissues, but no significant prognostic value was observed in the CBX5-high group.

CBX2 was identified as being overexpressed in ovarian cancer, BC, CRC and osteosarcoma, and high CBX2 levels predicted unfavorable outcomes in ovarian cancer and BC [20,44,45,46]. Serving as an oncogene, CBX2 promoted cell proliferation and invasion in CRC [20] and osteosarcoma [46]. In addition, the knockdown of CBX2 reduced anchorage-independent proliferation and enhanced anoikis-dependent apoptosis in ovarian cancer [44]. In our study, CBX2 was found to be upregulated in EC compared to normal tissues, while no statistical significance was observed.

CBX4 was reported to be elevated in various cancers, including lung cancer [19], osteosarcoma [47] and BC [48], and might serve as a novel therapeutic biomarker in clinical treatment. Wang et al. found that by transcriptionally upregulating Runx2 via recruiting GCN5 to the Runx2 promoter, high CBX4 promoted metastasis in osteosarcoma and might be a potential therapeutic target in metastatic osteosarcoma [47]. CBX4 also could be an index of drug resistance. Zhao et al. reported that CBX4 was overexpressed in HCC tissues, and blocking CBX4 expression could enhance the ability to overcome sorafenib resistance in HCC treatment [49]. In our study, high expression of CBX4 was related to poor OS in ESCC patients. CBX4 was also associated with lymph node metastasis, suggesting that CBX4 might involve in the progression of EC. 

As reported by previous research, CBX6 was downregulated in BC and negatively regulated by EZH2. Additionally, high expression of CBX6 inhibited cell invasion and proliferation of BC cells in vitro [50]. Another study indicated that downregulation of CBX6 induced MMP-2 expression and an invasive phenotype in malignant mesothelioma cells [51]. In our study, CBX6 was observed to be associated with different immune cells’ infiltration and might be involved in the immunotherapy of EC.

As for CBX7, it is the most well-studied protein among CBXs in cancer. Prior studies have demonstrated that CBX7 exerts tumor-suppressing properties in most malignancies. According to prior studies, CBX7 was downregulated in colon cancer [52], pancreatic cancer [53], thyroid cancer [54] and BC [55]. Downregulation of CBX7 indicated more aggressive phenotypes and worse survival in pancreatic cancer and thyroid cancer [53,54]. Different signaling pathways, such as PTEN/AKT signal transduction and Wnt/β-catenin pathway, might be involved in these tumor suppression courses [56,57]. On the contrary, some studies have revealed that CBX7 was overexpressed in prostate cancer [58], GC [15] and lymphoma [59]. The upregulation of CBX7 is related to age, lymph node metastasis and clinical stage, and indicates worse prognosis in GC patients [15]. In the present study, although low expression of CBX7 in EC compared to normal tissues was observed, no statistically significant differences were detected. Nevertheless, tumors of N3 nodal metastasis harbored lower CBX7 levels compared to N1 and N2 samples. Meanwhile, a lower expression of CBX7 predicted poorer OS in EAC patients. Collectively, CBX7 might have anticancer effects in EC, and the specific mechanism needs to be further explored. 

CBX8 plays paradoxical roles in multiple tumors. Tang’s study revealed that CBX8 promoted the proliferation but inhibited the migration, invasion and metastasis of CRC in vitro and in vivo, exerting paradoxical effects in CRC progression [60]. In terms of ESCC, CBX8 was demonstrated to promote cell proliferation and invasion [61] but suppress EMT and tumor metastasis by directly binding to the Snail promoter [62]. Our study found that high expression of CBX8 was associated with unfavorable OS in ESCC, which might be attributed to its tumor promotion effect.

The tumor microenvironment (TME), composed of all of the components of a solid tumor that are non-malignant cells, such as endothelial cells, CAFs, the extracellular matrix, inflammatory cells and all types of immune cells, has been an overarching focus in cancer studies during the past few decades [63]. Via complex crosstalk with tumor cells, the TME is involved in cancer evolution and progression [64]. Furthermore, the TME can facilitate drug resistance by decreasing drug penetration, maintaining the proliferative and anti-apoptotic abilities of surviving tumor cells [65]. As a critical component of TME, immune cells can release growth factors and cytokines to affect tumor development with either pro-tumorigenic or anti-tumorigenic properties. Previous studies have found that CBXs were related to the infiltration level of immune cells in CRC and sarcoma, such as CD8+ T cells, CD4+ T cells, CAFs, myeloid dendritic cells, macrophages and B cells, indicating that CBXs might be involved in the regulation of the TME [66,67]. In the current study, we found that CBXs were related to the infiltration of multiple immune cells and immune regulators, indicating that they might be involved in immunomodulation and immunotherapy in EC, which needs to be further explored.

There were some limitations that should be noted in the present study. First, all the data analyzed in our study were obtained from online public databases, and further prospective trials are warranted to validate the results. Second, further experiments are needed to clarify the underlying mechanisms of CBX3 in EC. Finally, clinical studies might be required to investigate the clinical applications of CBXs.

## 5. Conclusions

Our study comprehensively investigated the expression pattern, prognostic value and potential biological functions of CBX family proteins in EC. We found that CBX3 and CBX5 were upregulated in EC compared to normal tissues. In addition, we demonstrated that CBXs were related to various immune cells in EC. Additionally, we discovered that a high expression of CBX3 accelerated the malignant phenotypes of EC cells in vitro and that CBX3 might be an important oncogene involved in EC progression. However, further prospective clinical trials are warranted to confirm these results, and experiments are needed to uncover the specific mechanism of CBX3 in EC progression.

## Figures and Tables

**Figure 1 genes-13-01582-f001:**
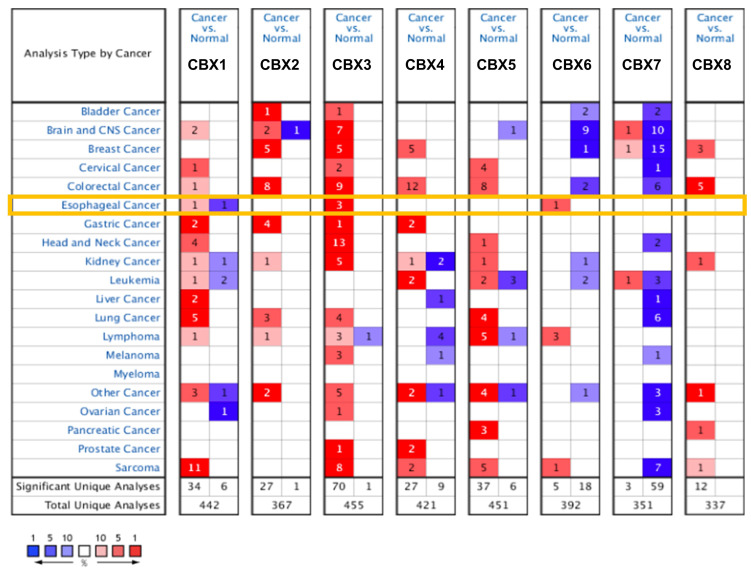
Transcriptional expression of CBX family proteins in 20 types of cancers from Oncomine. As shown in the orange frame, CBX3 and CBX6 were significantly upregulated in EC tissues compared to normal tissues. The number in the graphic represents the numbers of datasets with statistically significant changes in the mRNA expression of the target gene: upregulated (red) and downregulated (blue), and the darker the color, the higher or lower the expression. The criteria used was as follows: *p*-value: 0.01, fold change: 2, gene rank: 10%, data type: mRNA, analysis type: cancer vs. normal tissue.

**Figure 2 genes-13-01582-f002:**
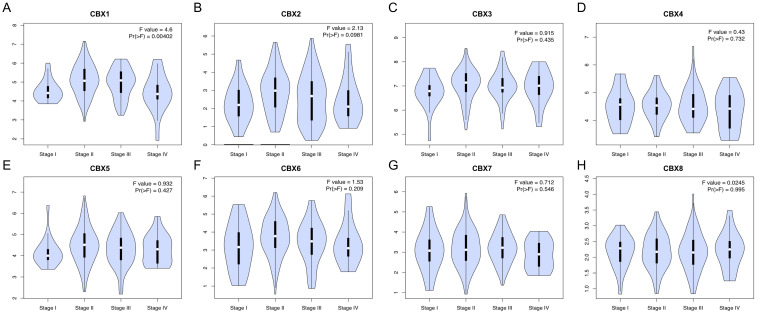
Relationships between the expression of CBX family proteins and clinical stages of EC (**A**–**H**). (GEPIA2, *p* < 0.05 was considered statistically significant). CBX1 expression was related to pathological stages of EC patients.

**Figure 3 genes-13-01582-f003:**
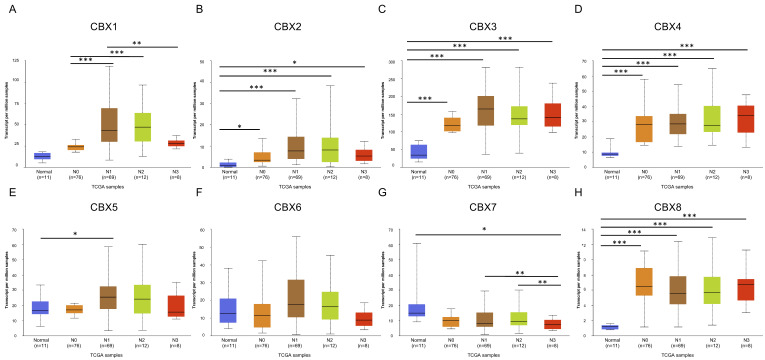
Relationships between the expression of CBX family proteins and nodal metastatic status in EC patients (**A**–**H**). (UALCAN, * *p* < 0.05, ** *p* < 0.01, *** *p* < 0.001).

**Figure 4 genes-13-01582-f004:**
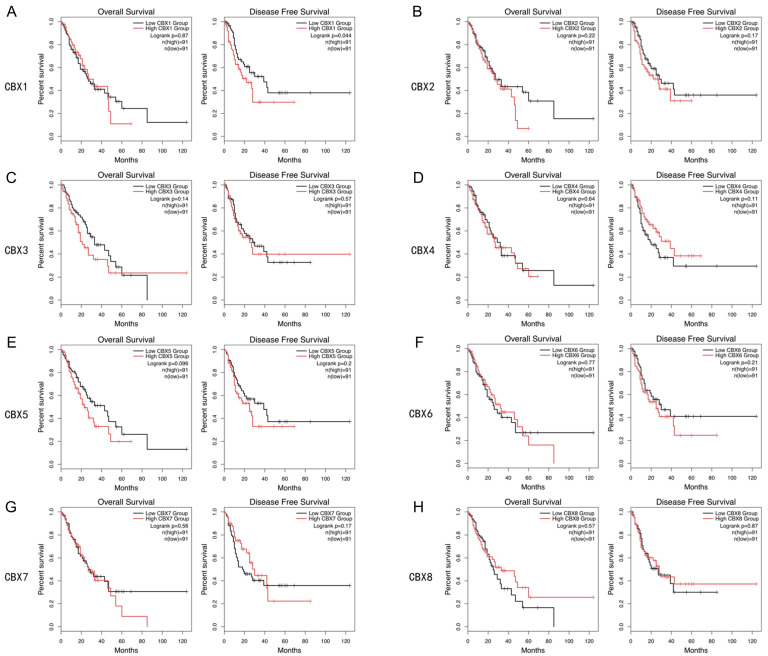
Prognostic value of CBX family proteins in EC patients (**A**–**H**). (GEPIA2, *p* < 0.05 was considered statistically significant).

**Figure 5 genes-13-01582-f005:**
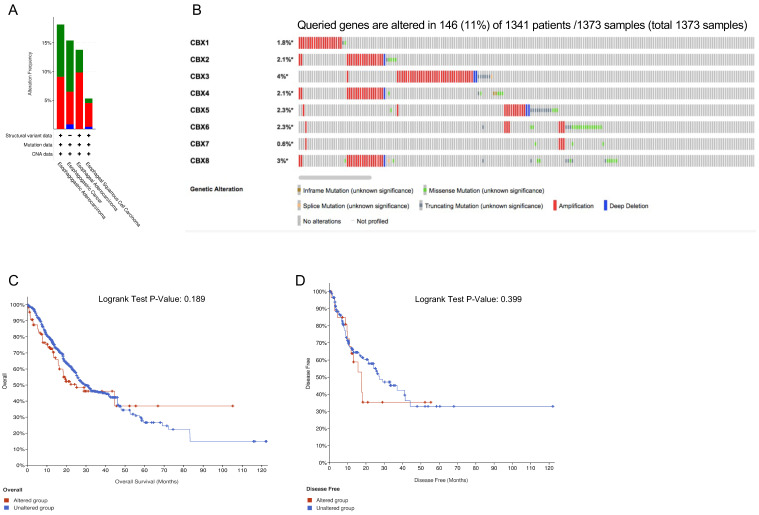
Genetic alterations of CBX family proteins and the related prognostic value in EC. (**A**) Summary of CBXs alterations in EC; (**B**) genetic alterations of CBX family proteins in EC; (**C**) relationship between overall survival (OS) and CBX genetic alterations; (**D**) relationship between disease free survival (DFS) and CBXs genetic alterations. (cBioPortal, *p <* 0.05 was considered statistically significant).

**Figure 6 genes-13-01582-f006:**
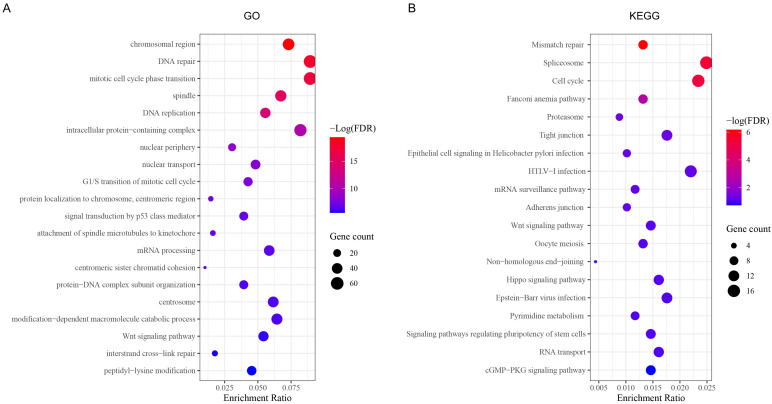
GO and KEGG analysis of CBX family proteins in EC. (**A**) Gene ontology (GO) enrichment of CBXs in EC; (**B**) Kyoto Encyclopedia of Genes and Genomes (KEGG) pathways of CBXs in EC.

**Figure 7 genes-13-01582-f007:**
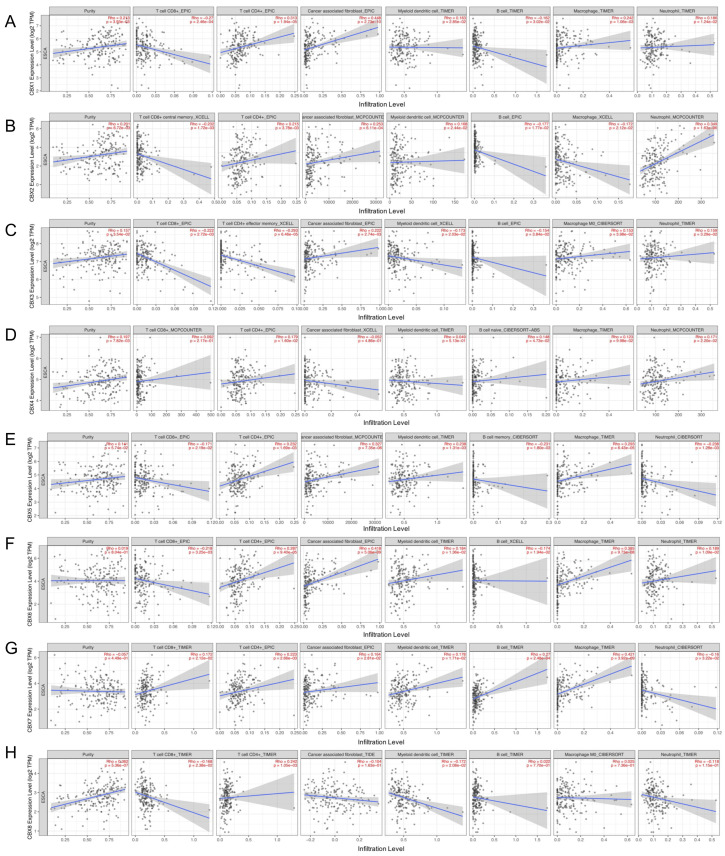
Relationships between CBX1-8 and immune cells infiltration level (**A**–**H**). (TIMER2, Rho > 0, positively correlated; Rho < 0, negatively correlated).

**Figure 8 genes-13-01582-f008:**
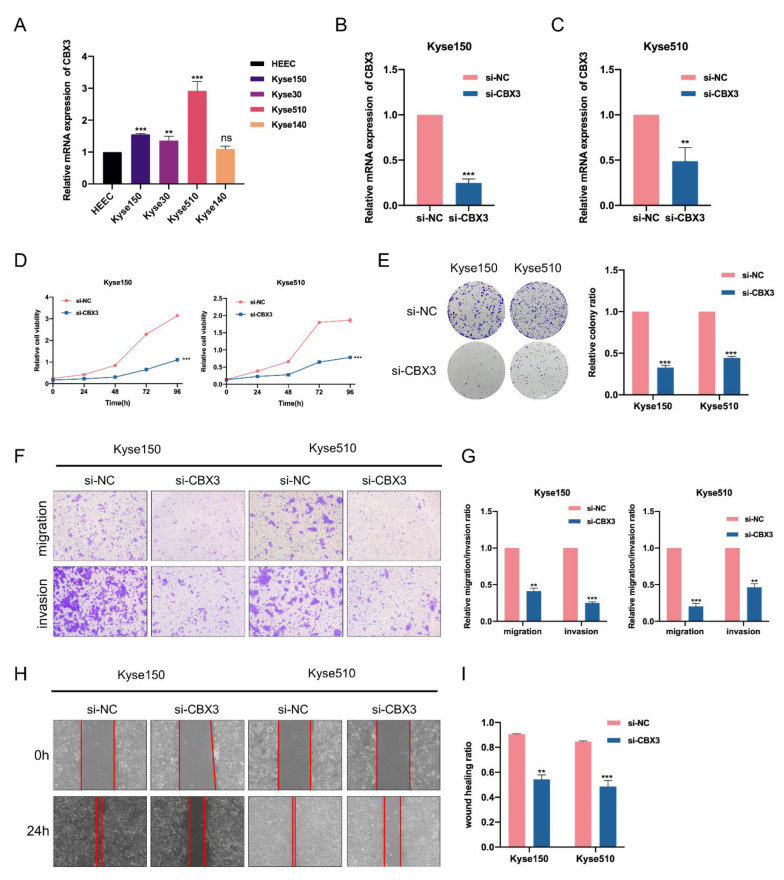
Knockdown of CBX3 suppressed the proliferation, migration and invasion of EC cells. (**A**) RT-q-PCR was used to evaluate the expression of CBX3 in EC cell lines compared to the normal esophageal epithelial cell line HEEC. (**B**,**C**) The knockdown efficiency of CBX3 in EC cell lines was verified by RT-q-PCR. (**D**,**E**) CCK8 and colony formation assays were applied to evaluate the effect of CBX3 depletion on proliferation of EC cells. (**F**,**G**) Effects of CBX3 silencing on EC cells migration and invasion were assessed by means of transwell assay. (**H**,**I**) Wound-healing assay was conducted to evaluate the effect of CBX3 silencing on migration of EC cells. Data are expressed as the means ± SD for *n* = 3. ** *p* < 0.01; *** *p* < 0.001.

## Data Availability

All data applied in the study are available through the online website offered in the Section 2.

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
