# Peer review of "Expression and Prognostic Value of Chromobox Family Proteins in Esophageal Cancer"

_genes, 2022, doi:10.3390/genes13091582_

Round 1

Reviewer 1 Report

Dear Editors,

Dear Authors,

I read with interest the manuscript entitled “Expression and prognostic value of Chromobox family mem-

bers in esophageal cancer”.  The authors used several bioinformatics analyses softwares to investigate the expression profiles and prognostic role of proteins belonging to the Chromobox (CBX) family.  The authors found that some CBX family proteins were overexpressed in esophageal cancers, while others were associated to specific tumors or with worse disease-free survival. Overall, the manuscript is well written, the methodology is appropriate and scientifically sound. However, I do have some major comments that definitely need to be addressed.

1.       Major English editing is necessary throughout the manuscript. Some verbal tenses are not appropriately used. Additionally, special attention should be put on abstract and conclusion.

2.       In the title, the word “members” should be changed with the word “proteins”. That is because the current title could be misleading for the reader that is not aware of the CBX family.  

  1. The introduction appropriately puts this research in the context of other studies. However, the authors should also report and cite a recent narrative review which comprehensively evaluated novel strategies for the diagnosis of esophageal cancer via blood biomarkers, including proteins and miRNAs. (Modern Diagnosis of Early Esophageal Cancer: From Blood Biomarkers to Advanced Endoscopy and Artificial Intelligence. Cancers, 13(13), 3162. https://doi.org/10.3390/cancers13133162)

4.       Page 2, lines 71-72: “might involve”, did the authors mean “might play”?

  1. The conclusion should speculate on possible further research that should follow this present study. Additionally, major English polishing is needed here. As for the English, I would suggest:  Our study comprehensively demonstrated investigated the expression pattern, prognostic value and potential biological functions of CBX family members in EC. The relationships between CBXs and immune cells and immune regulators were revealed explored as well. Whats more Additionally, we discovered that CBX3 might be an important oncogene involved in EC progression.

6.       The manuscript is somewhat overcrowded with small and low-quality figures, making it very confusing to read. Some of the figures are definitely too small and unreadable. I would suggest reducing the number of figures in the manuscript. Only figures showing statistically significant results should be kept (in bigger and higher quality format) (Other figures should be uploaded as supplementary material).

Reviewer 2 Report

In the present review entitled “Expression and prognostic value of Chromobox family members in esophageal cancer”, the authors aimed to assess the impact of chromobox family expression on overall survival and relapse-free survival in patients with esophageal cancer. The study has limitations, and some questions and considerations need to be reviewed and answered by the authors.

1.       Lines 40-42 the authors said: “Based on the difference of anatomic sites, esophageal squamous cell carcinoma (ESCC) and esophageal adenocarcinoma (EAC) are the 2 most common histological subtypes”.

Authors should rewrite this sentence, as the sentence does not seem correct, as both subtypes affect the same anatomical region, differ in risk factors and incidence trends, and are pathologically distinct.

1.       Lines 43-45 the authors said: “Targeted therapy and immunotherapy also give rise to survival rate of EC patients with the rapid development and more mature recognition of molecular biomedicine and immunomedicine.”

The authors should clarify that these immunotherapy studies are phase II or III.

2.       Line 59: “…, belong to HP1 group...”

Please, write the HP1 name: Heterochromatin protein 1.

3.       Lines 128-129: “Human esophageal cancer cell lines Kyse150, Kyse510, Kyse30, Kyse140 were obtained from the Shanghai Cell Bank of Chinese Academy of Sciences (Shanghai, China)”.

Are these selected cells all from esophageal squamous cell carcinoma? Please put this information in the text.

4.       Lines 174-176: “As shown in Figure 1, CBXs were differently expressed in 20 various types of cancers. For EC, CBX3 was overexpressed in tumor tissues, while the expression of other CBX family members was not statistically significant.”

Please explain these results because they are unclear: which tumors have significantly overexpressed CBX genes and which are these? What is the meaning of the numbers in the boxes in Figure 1 (red and blue)? I suggest rewriting the legend in Figure 1, making it more explicit.

5.       Lines 188-190 (Figure 2):

 Please review the subtitle.

6.       Line 208: “Figure 3. Correlations between mRNA expression.”.  

Is this analysis of correlation or association?

7.       Lines 298-299, the authors said: “Based on the above analysis, we speculated that CBX3 might act as an oncogene in EC progression.”

 What were the results on which the authors relied to write this statement? It was unclear why the authors chose this gene for the next experiments.

8.       Lines 300-302: “We found 300 that CBX3 was up-regulated in the three EC cell lines (Kyse150, Kyse510 and Kyse30) com- 301 pared to normal esophageal epithelial cell HEEC (Figure 12A)”

The authors did not mention the normal esophageal cell in the methodology (HEEC), please add this information to 2.9 subitem and describe this cell lines information.

9.       Why did the authors choose only esophageal squamous carcinoma cell lines for the CBX3 siRNA experiments? I asked it because CDX3 appears more important to EAC  (lines 324-325: “We found that CBX3 and CBX5 were upregulated in EC, and high CBX3 predicted shorter OS in EAC “) and the authors used esophageal squamous cell lines to experiments.

10.   Why did the authors not perform all analyzes by histological subtype (ESCC and ADC)? It is known that they are diseases with different etiological factors and the tumors' biology. I recommend that the authors perform these analyses by separating the subtypes and putting this data as supplementary material.

11.   I suggest that the authors perform western blotting to assess whether the protein is silenced, as is the mRNA.

12.   The limitations of this study should be described in discussion.

13.   The discussion about all CBXs is very extensive. On the other hand, I miss further discussing the CBX3 gene, how it is expressed in other tumors, its association with clinicopathological data, talking about possible pathways and genes that can be regulated or are regulating this gene.

14.   Lines 427-428 “What’s more, we discovered that CBX3 might be an important oncogene involved in EC progression.”

Please, explain this sentence. How did the autho

Reviewer 3 Report

I've read the article "Expression and prognostic value of Chromobox family members in esophageal cancer" with interest. It represents an important study aimed to investigate which chromobox (CBX) family proteins might be potential biomarkers in esophageal cancer carcinogenesis and development. The experiment design and the presentation of data are excellent. The manuscript was well-constructed and written. There is a clear effort in the work. I congratulate the authors. However, one point should be addressed by the authors.

Comment:

1. For all or most experimental techniques that have been described in the MM section, it is suggested to use a relevant reference.

Round 2

Reviewer 2 Report

The authors answered the questions and performed the requested experiments, so I consider the manuscript ready for publication.